# Recurrent Context Compression: Efficiently Expanding the Context Window of LLM

## Abstract

To extend the context length of Transformer-based large language models (LLMs) and improve comprehension capabilities, researchers often encounter constraints stemming from finite computational resources and bounded memory capacities. This work proposes a novel approach, termed Recurrent Context Compression (RCC), designed to efficiently expand the context window length of LLMs. Furthermore, we delve into the prevalent issue of degraded model performance when both instructional prompts and contextual information undergo compression for downstream tasks. To address this challenge, we propose a novel instruction reconstruction methodology aimed at mitigating the detrimental effects of this compression process. The effectiveness of our proposed approach was validated across multiple tasks while achieving an impressive context compression rate of at least $32\times$. On text reconstruction task, we maintain a BLEU-4 score close to 0.95. On passkey retrieval task, we achieve nearly 100% accuracy involving an extensive sequence length of 1 million tokens. On long-text question-answering task, we obtain comparable performance with the non-compressed LLM in F1 and Rouge scores. Our method also demonstrated competitive performance in long-text question-answering tasks compared to non-compressed methods, while significantly saving storage resources.

## 1 Introduction

With the rapid advancement of natural language processing technologies, Transformer-based large language models (LLMs) have become a key driving force in this field. However, when grappling with lengthy text inputs that extend beyond a certain scope, LLMs frequently encounter limitations imposed by their finite context window length capabilities. These limitations stem from several inherent factors in the model architecture and training methods. Firstly, during the inference phase, models are constrained by the pretraining text length, leading to a significant decline in quality when the generated sequence exceeds the pretrained context window. Secondly, the design of the Transformer architecture requires storing information from the entire input sequence, which leads to a significant memory usage from the KV-Cache during inference. To address these issues, related works have extended the context window of LLMs (Hochreiter & Schmidhuber, 1997; Child et al., 2019; Wu et al., 2022; Rae et al., 2019; Bulatov et al., 2022; Liu et al., 2023; Mohtashami & Jaggi, 2023; Beltagy et al., 2020) by optimizing training methods, model structures, and context compression. Among these, context compression techniques (Rae et al., 2019; Snell et al., 2022; Chevalier et al., 2023; Wingate et al., 2022; Mu et al., 2023; Ge et al., 2023; Munkhdalai et al., 2024; Ren et al., 2023; Li et al., 2023) are popular promising because they reduce the length of contexts or prompts while maintaining performance. This allows for the inference of longer context windows within limited resources. Figure 1 compares the memory resource consumption of our method with non-compression methods. Additionally, most compression-based methods can be integrated with other context window extension techniques to further enhance performance.

However, existing context compression methods face with three major challenges in long-text language modeling. Firstly, The current SOTA model, ICAE (Ge et al., 2023), can only achieve up to $8\times$ compression; beyond that, performance significantly declines. Secondly, most compression methods still conduct experiments within a 4k context range, which is unable to keep pace with the development of current large language models (LLMs). For example, GPT-4 has reached a context length of 128k. Therefore, it is crucial to continue extending the context length further to adapt to

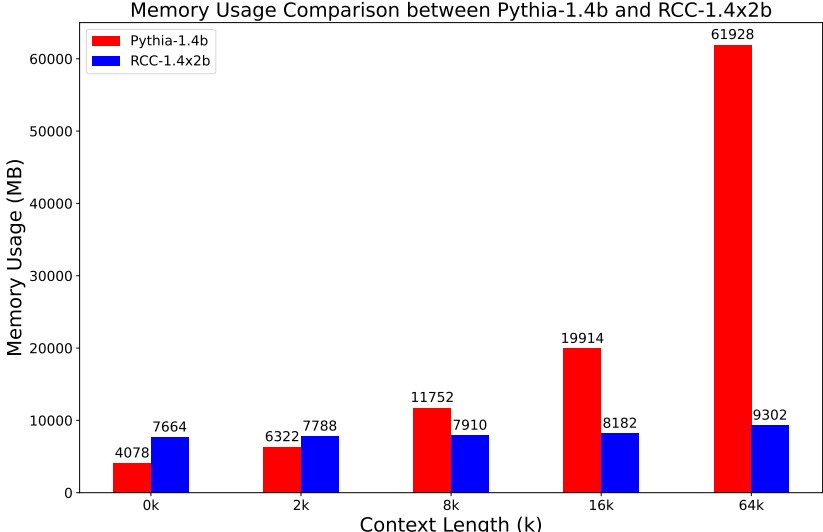

Figure 1: GPU memory Consumption of Different Models with Increasing Length. Left: Pythia-1.4b, Right: RCC model using Pythia-1.4b for both encoder and decoder. Both models utilize FlashAttention-2 (Dao, 2023). A more detailed analysis of GPU memory consumption can be found in Appendix A.

these advancements. Lastly, we observed that context-compressed language models face the issue of context-instruction confusion in downstream tasks. When both context and instructions are compressed simultaneously, the model often struggles to follow instructions correctly, resulting in lower performance, as show in Figure 2. This issue has not been addressed in previous studies.

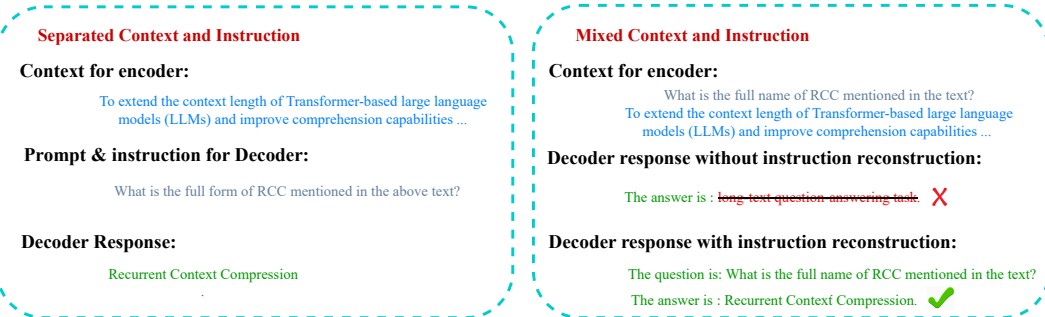

Figure 2: RCC encoder first compressing a long context into short span of vectors. Then the compressed vectors can be conditioned on by the RCC decoder to respond to various prompts. RCC can deal with complex context with instruction mixed in (Right), while existing methods can only handle the separated case (Left).

To address the aforementioned issues, this paper makes the following contributions:

Firstly, we propose a context compression model, called Recurrent Context Compression (RCC) architecture. RCC significantly reduces information loss during the compression process, greatly enhancing compression efficiency. In experiments, we achieved nearly 0.95 BLEU-4 scores with a $32\times$ compression rate on text reconstruction tasks, as shown in Figure 4.

Secondly, we propose a recurrent compression mechanism to extend the model's ability to compress texts beyond the encoder's window length. When training long-text context compression language models, the length of the context sequence input to the encoder is proportional to the required computational resources, limiting the extension of context length during training. Therefore, we use a simple yet effective solution: initially, conducting full-parameter training on shorter sequences.

Subsequently, we freeze the encoder and continue training on longer sequences, enabling the extension of training context length . Our model exhibits excellent extrapolation ability during the inference stage, with the capability to further extend lengths up to 1M. For more details, please refer to Section 4.2

Lastly, in downstream text generation tasks, we found that when both context and instruction are compressed, the model is hard to follow instructions, leading to a decline in response quality. To mitigate this issue, we leverage the text reconstruction capability of the context compression language model, allowing the decoder to reconstruct the instruction from the compressed vectors and continue generating responses based on the instructions. This significantly improves the output quality for mixed context and instructions, achieving results close to those obtained by inputting instructions directly into the encoder.

## 2    RELATED WORK

**Context Compression:** Early approaches to context compression aimed to derive sentence representation vectors for tasks such as document retrieval. Transformer-based autoencoder architectures like TSDAE (Wang et al., 2021) and Nugget (Qin & Van Durme, 2023) are relevant to our work. In TSDAE, noise such as word deletion or swapping is added to input sentences to train sentence embedding vectors. The encoder compresses the corrupted sentence into a fixed-size vector, which the decoder then reconstructs into the original input text. However, such approaches cannot be directly applied to text generation tasks. Git(Mu et al., 2023) compresses prompts by fine-tuning LLM. During training, a specific masking matrix compresses prompts into a few Gist tokens, which can still prompt the language model for responses. Similar prompt compression work was proposed by (Wingate et al., 2022). However, these tasks only compress prompts.

Several works (Snell et al., 2022; Chevalier et al., 2023; Ge et al., 2023; Ren et al., 2023; Li et al., 2023; Jiang et al., 2023b; Munkhdalai et al., 2024) have focused on context compression. ICAE(Ge et al., 2023) is similar to our work but suffers from lower compression efficiency and lacks extensive research on longer sequences. Additionally, AutoCompressors(Chevalier et al., 2023), which recursively compress long texts into summary vectors to extend context length, are also relevant to our approach. Selective context, proposed by (Li et al., 2023), identifies and prunes redundancy in the input context to enhance LLM inference efficiency, making inputs more compact. Recently, (Munkhdalai et al., 2024) proposed a similar work combining two attention mechanisms with context compression functionality, showing promising results. However, this new attention mechanism cannot be directly applied to pre-trained open-source LLMs and faces attention optimization challenges in practical applications. None of the studies have thoroughly investigated the problem of instructional confusion that arises when both instructions and contextual information are subjected to compression. Our work introduces a new solution to mitigate this issue.

**Long Context LLM:** LLMs typically fix the context window length during training, such as the Pythia (Biderman et al., 2023), LLaMA (Touvron et al., 2023a;b), and Mistral (Jiang et al., 2023a) series. Consequently, researchers have explored various methods to extend the context window length of pre-trained language models. These methods(Chen et al., 2024; Tworkowski et al., 2023; Chen et al., 2023; Liu et al., 2023; 2024; Yen et al., 2024), which have achieved notable results based on existing pre-trained models. Our approach can be combined with these methods, applying them to either the encoder or the decoder to achieve more extended compression effects.

Additionally, our work is inspired by language models with recurrent structures (Hochreiter & Schmidhuber, 1997; Gu & Dao, 2023; Sun et al., 2023; Peng et al., 2023). These models compress historical context within a certain range into the hidden state of a single time step, enabling the current token to access information from the previous step for inference. They demonstrate strong competitiveness with Transformer models, indicating that compressing token information over a certain length can achieve lossless inference.

## 3 DESIGN OF RCC

### 3.1 RCC MODEL ARCHITECTURE

As shown in Figure 3, the RCC model architecture is similar to ICAE(Ge et al., 2023), consisting of an encoder and a decoder. In connecting the encoder and decoder, RCC differs from ICAE and AutoCompressors(Chevalier et al., 2023). While ICAE and AutoCompressors uses the final layer's vectors from the encoder as input for the decoder, we utilize the output information from each layer of the encoder. This information is then using MLP layers and inputed into the decoder. This method obtains richer feature information. The encoder can be a Transformer-based LLM or an RNN-based LLM, while the decoder is a Transformer-based LLM. The encoder is responsible for compressing the information which is utilized by the decoder for inference. The decoder can fully learn the compressed information vector at any position using the attention mechanism. After training, the maximum context length of RCC is the decoder context window length multiplied by the encoder compression rate, e.g., $1M = 2k \times 512$.

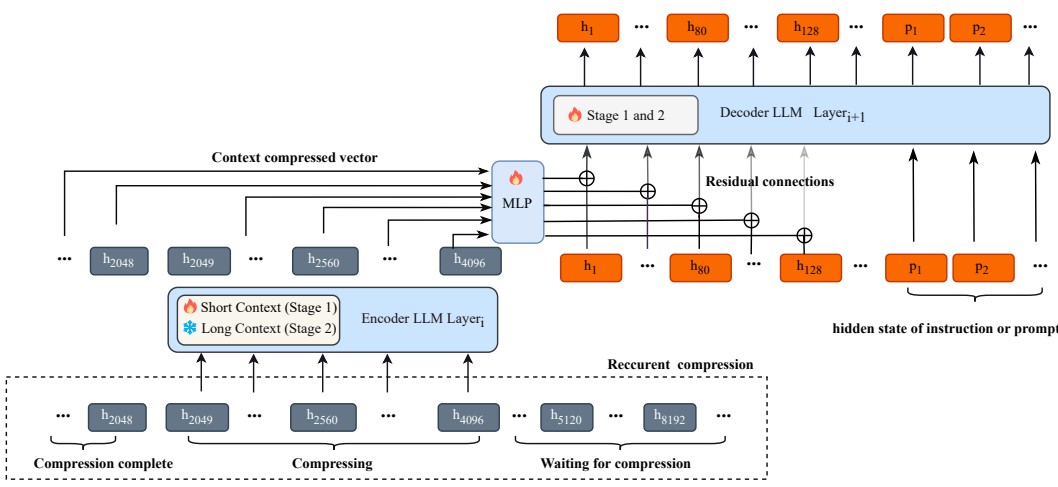

Figure 3: The structure of the encoder and decoder in RCC layer i. We segment the long sequence and then iteratively compress each segment. We take the hidden layer vector corresponding to the last token of each short sequence within a segment as the compression vector. We employ a two-stage training method. In the first stage, we train on short sequences across all parameters. In the second stage, we input long sequences and freeze the encoder's parameters, training only the MLP and decoder parameters.

### 3.1.1 RCC ENCODER

The design of RCC's encoder is inspired by the Mamba-based LLM (Gu & Dao, 2023). Mamba is essentially a state space model, similar to an RNN. In Mamba, the current token only needs to access the state vector from the previous timestep to complete the current inference step. However, as the context length increases, the performance of Mamba deteriorates. This indicates that the state vector at each time step in Mamba can store only a limited length of historical context information. Therefore, we propose a compromise: for long sequences, we can divide them into fixed-length short sequences and iteratively compress each short sequence into a state vector. Finally, we concatenate the state vectors of each short sequence as the historical state information to be input into the decoder. This approach retains complete historical information while leveraging the model's compression capabilities to save memory. This divide and iteratively compress strategy can also be combined with transformers. Our experiments show that Transformers also have this capability because a Transformer can be viewed as a special state space model or RNN (Feng et al., 2024).

The primary task of the RCC encoder is to compress long sequences. The initialized encoder is a pretrained language model, which can be based on either Mamba or Transformer architectures. By setting a compression rate, we divide long sequences into fixed-length short sequences and iteratively feed these short sequences into the encoder. As illustrated in Figure 3, we find the last token

of each short sequence, and use these tokens' output vectors from each layer as compression vectors. We concatenate these compression vectors, pass them through a linear layer, and finally input them into the corresponding layer of the decoder. Through this method, the RCC's encoder accomplishes the compression modeling of the entire long context.

### 3.1.2 RCC DECODER

The decoder of the RCC is a Transformer-based language model responsible for the final text inference. Its inputs include the compressed vectors from the encoder and token embedding vectors related to the prompts. Each layer's compressed vector from the encoder passes through a linear layer before being input into the decoder. For the first layer's mapped vector, we concatenate it with the decoder's token embedding vector and then feed it into the first layer of the decoder. Subsequent, each encoder layer's output is connected to the corresponding decoder layer's compressed vector. This connection is established through residual connections, as shown in Figure 3. It is crucial to note that only the output vectors corresponding to the compressed information will have residual connections, while the other output parts remain unchanged. If the number of compressed vector layers does not match the number of decoder layers, we apply simple rule-based mappings, either by duplicating to increase the number of layers or averaging to reduce the number of layers.

### 3.2 MODEL TRAINING

#### 3.2.1 TWO-STAGE APPROACH

We use an iterative segmented computation method to process long sequences. During the model inference phase, we only need to store the compressed vectors, so memory limitations do not become a bottleneck for inferring long texts. However, during the training phase, we also need to store the gradient information of the entire long sequence, which significantly exceeds the memory limit. We mitigate this issue using a simple yet effective two-stage training method. In the first stage, we perform full-parameter fine-tuning with a large number of shorter sequences, allowing the encoder to sufficiently learn how to compress the context into a single vector. Correspondingly, the decoder learns to infer or reconstruct from the compressed vectors. After the first stage of training, the encoder can produce stable compressed vectors. At this point, we input longer sequences for training while freezing the encoder to save memory resources. This approach not only reduces training costs but also enables the decoder to learn how to handle a larger number of compressed vectors. This method does not require complex gradient optimization algorithms or substantial GPU memory resources(Liu et al., 2023; Chevalier et al., 2023; Liu et al., 2023) and can efficiently scale to longer sequences. We validated the effectiveness of this method on a 1M-length key retrieval task.

#### 3.2.2 RECONSTRUCTION AND CONTINUATION TASKS

For the model training tasks, we require the model to possess both contextual memory and contextual reasoning abilities. Therefore, we select text reconstruction and text continuation tasks. Traditional autoencoding text reconstruction tasks (Ge et al., 2023) can only perform holistic reconstruction in a fixed manner, making it impossible to handle long texts. To address this issue, we propose another new training task called random prompt text reconstruction task. Specifically, we randomly extract a short text segment from the encoder's input text as a prompt to the decoder, requiring the decoder to reconstruct the content following the prompt by leveraging the compressed information and the prompt. This task enhances the model's memory and retrieval ability. Additionally, to maintain the model's reasoning ability, we employ text continuation tasks. Relevant formulas can be found in Appendix B.

#### 3.2.3 INSTRUCTION-AWARE RECONSTRUCTION

Through our experiments, we have discovered that when instructions and context are mixed, they are treated equally by the compression model. This leads to the model struggling to follow instructions effectively, resulting in a significant drop in performance. To address the problem we propose an instruction reconstruction method. During the training phase, we input the instruction as part of the context to the model's encoder, with the instruction randomly placed at the beginning or end of the

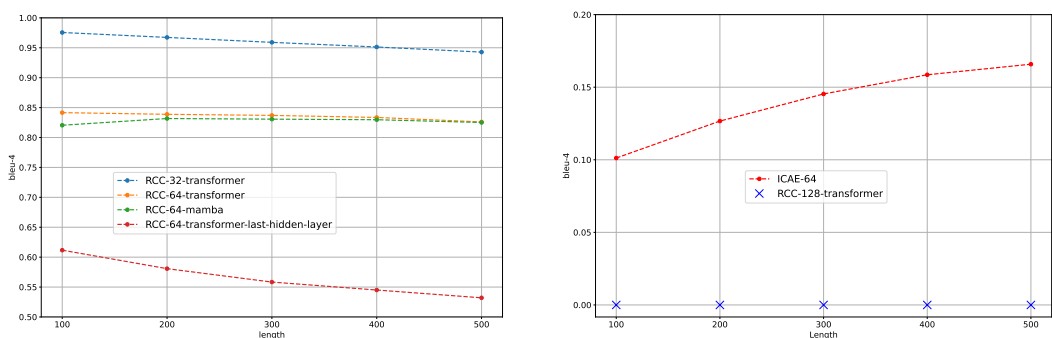

Figure 4: Text Reconstruction Scores of Different Models.

context. The decoder is then required to firstly reconstruct the instruction and subsequently answer the question based on the instruction.

## 4 EXPERIMENTS

First, we conduct text compression rate experiments using the random prompt text reconstruction task, selecting ICAE (Ge et al., 2023) as the baseline model. Subsequently, we evaluate our method's performance on long text tasks, including the passkey retrieval task(Mohtashami & Jaggi, 2023) with 1M sequence length and the long document question-answering benchmark in Longbench (Bai et al., 2023). For model architecture, we test the effectiveness of Pythia-1.4b ((Biderman et al., 2023), Apache License 2.0) and Mamba-1.4b ((Gu & Dao, 2023), Apache License 2.0) as encoders. Both Pythia-1.4b and Mamba-1.4b only support a 2048 context window. The decoder was pythia-1.4b. We randomly sampled about 5 billion tokens from the pile ((Gao et al., 2020), MIT License) dataset as the training set, concatenating these tokens into a continuous ultra-long one-dimensional array. The learning rate was set to 1e-4. We only training for one epoch, typically completing within 30 hours on a server with 8 A800 GPUs.

### 4.1 TEXT RECONSTRUCTION

In the Pre-training phase, we use a combination of random prompt text reconstruction tasks and text continuation tasks, with a ratio of 9:1. Afterwards, we fine-tune the model using a small amount of random prompt text reconstruction tasks data. The input sequence length for the encoder is set to 2048 tokens, while the uncompressed part of the decoder had an input length of 512 tokens. For the random prompt text reconstruction task, the decoder needs to reconstruct the content following a given prompt. The prompt and the content to be reconstructed are a subsequence of the compressed sequence. The prompts begin this subsequence and are excluded from the loss calculation during training. The loss was calculated only for the reconstructed text following the prompts. In the text continuation task, the decoder's uncompressed input sequence was the continuation of the compressed sequence. We evaluated the model's compression performance using the BLEU-4 score, comparing the reconstructed text to the actual text. We created 100 encoder input samples, each with a token length of 2048. To ensure fairness in the scores, we selected 5 text segments as prompts for the decoder at every 300-token interval from the sample. Each decoder then calculated the reconstruction score for prompts at 5 different positions, and we averaged these scores. The prompt text length was about 10 tokens, and the reconstruction text length was about 500 tokens. Reconstruction examples can be found in Appendix D.

As shown in Figure 4, we compared different models under a 64× compression rate. Both RCC-64-mamba and RCC-64-transformer achieved a BLEU-4 score close to 0.8, but mamba's training time was nearly 1.5 times that of transformer. RCC-64-transformer-last-hidden-layer, which uses only the encoder's last layer compression vectors, achieved a BLEU-4 score of approximately 0.6. This approach, common in traditional autoencoder models (Wang et al., 2021; Qin & Van Durme, 2023; Ge et al., 2023), retains less textual information compared to using compression vectors from all layers. Additionally, ICAE performed poorly under a 64× compression rate, with a BLEU-4 score

of about 0.1, confirming our method's effectiveness in preserving text information. At the same time, we tested reconstruction performance under different compression rates. At a 32× compression rate, the BLEU-4 score reached 0.95. At a 64× compression rate, the score dropped to between 0.8 and 0.85. At a 128× compression rate, We encountered an issue where the loss failed to converge, preventing BLEU-4 score computation. This indicates that higher compression rates increase the difficulty of text reconstruction.

| Model | 32K | 128K | 256K | 512K | 1M |
|---|---|---|---|---|---|
| Infini-Transformer-FT | 100/100/100 | 100/100/100 | 100/100/100 | 97/99/100 | 96/94/100 |
| RCC-Mamba-FT-8k | 98/100/97 | 96/98/94 | 95/93/89 | 94/95/96 | 94/96/96 |
| RCC-Transformer-FT-8k | 97/95/96 | 96/97/96 | 92/96/96 | 92/89/95 | 97/96/96 |
| **RCC-Mamba-FT-32k** | 100/100/100 | 100/100/100 | 100/100/100 | 100/100/100 | 100/99/100 |
| **RCC-Transformer-FT-32k** | 99/100/100 | 100/100/100 | 100/100/100 | 98/100/100 | 100/100/100 |

Table 1: The performance of the different models on the passkey retrieval tasks ranging from 32k to 1M sequence lengths, RCC-512-FT-8k denotes that the RCC model is trained with full parameters on a fine-tuning dataset with a length of 8k. RCC-512-FT-64K is trained on a fine-tuning dataset with a length of 64K based on RCC-512-FT-8k, while in this case, we freeze the encoder.

## 4.2 PASSKEY RETRIEVAL TASK

We utilize the passkey retrieval task (Mohtashami & Jaggi, 2023) to validate the effectiveness of the two-stage approach mentioned in section 3.2.1. Additionally, we observe that our method exhibits certain length extrapolation capabilities. This enables it to handle compressed vector lengths during inference that far exceed those seen during training. This indicates that the compressed vectors generated by our method can be reliably recognized by the encoder, with minimal influence from positional encoding. Passkey retrieval task involves embedding a random number into a long sequence composed of repeated fixed short phrases. The task requires the model to accurately retrieve the hidden number from these long sequences. Detailed construction methods for passkey retrieval task samples are provided in Appendix C. In this task, we employed a compression rate of $512\times$. Although this compression rate might not be effective for reconstruction tasks, experiments show that the fine-tuned RCC model performs well in the passkey retrieval task. The model was first pre-trained on a dataset containing only the random prompt text reconstruction task, with an encoder input length set to 8k and a non-compressed decoder input length of 512. After pre-training, we constructed nearly 30,000 passkey retrieval task samples with context lengths of 8k and 32k, respectively. These samples formed the fine-tuning dataset. We conducted a two-stage fine-tuning process. In the first stage, we fine-tuned the entire parameter set using the 8k context length samples. After completing the first stage, we proceeded to the second stage with the 32k context length samples. During this stage, we froze the encoder parameters to accommodate the constraints of limited available memory.

From Table 1, we can observe that even with the encoder using only an 8k context window. The model achieves almost 90% accuracy in passkey retrieval tasks up to 1M, demonstrating the strong length extrapolation capabilities of our model. After the second stage of fine-tuning with sequences up to 32k, the model achieves nearly 100% accuracy on passkey retrieval tasks up to 1M, proving the effectiveness of our two-stage training method, even with the encoder parameters frozen at this stage. Table 1 also shows that Compared to the model Infini-attention (Munkhdalai et al., 2024) which requires re-pretraining, our model has also achieved similar performance. Unlike Infini-attention, our method can be fine-tuned on existing open-source LLMs with a small amount of data, without requiring the reconstruction of the LLM model and pre-training with hundreds of billions of tokens.

## 4.3 LONG-TEXT BENCHMARK EVALUATION

### 4.3.1 EVALUATION DATASET

LongBench (Bai et al., 2023) is a benchmark designed to evaluate the capabilities of large language models in understanding long contexts. To evaluate our model's performance on texts of different

lengths, we selected the LongBench-E set. This dataset evenly covers test samples of various length ranges, allowing us to analyze the impact of length variation on performance. Due to limitations in the fine-tuning dataset, our work focuses on using the single-document QA and multi-document QA tasks for evaluation. These two document QA tasks consist of four subtasks, with each task containing between 150 to 300 samples. Detailed information on the evaluation dataset can be found in Appendix E.

### 4.3.2 INSTRUCTION FINE-TUNING

First, we conducted pretraining on the random prompt text reconstruction task and the text continuation task with a ratio of 1:9. We use two-stage approach for training, the first stage involves training with full parameters on texts with a length of 2k. In the second stage, the encoder is frozen, and training is conducted on texts with a length of 16k. For question answering instruction fine-tuning, we used the Prompt-with-Context (PwC) (Ge et al., 2023) and hotpotQA (Yang et al., 2018) datasets. These datasets include context with instructions and outputs, teaching the model to use context for answering questions rather than relying solely on internal knowledge. We concatenated the instructions and context as the encoder's input, while the instructions and output text were concatenated as the decoder's uncompressed input. We repeated the instructions twice to train the decoder to reconstruct instructions, enhancing mixed instruction and context effectiveness during inference. The PwC dataset has 240k samples, and hotpotQA has 90k samples. Additionally, to improve instruction reconstruction and maintain the decoder's instruction-following capability, we randomly selected 50k instruction samples from the orca dataset (Mukherjee et al., 2023). These samples lack explicit context fields and typically mix instructions with context. We set the context of the orca dataset to be empty and then processed it in the same way as PwC. During fine-tuning, the encoder's input context length was set to 2048 tokens. The uncompressed part of the decoder's input was specified as 512 tokens, while maintaining a compression rate of 32.

### 4.3.3 EVALUATION

We fine-tuned Pythia-1.4b with instruction pairs constructed from PwC, hotpotQA, and some orca data to ensure it follows instructions. We evaluated the fine-tuned Pythia-1.4b while our model using LongBench's(Bai et al., 2023) automated evaluation tools, covering two document QA tasks. More information related to the evaluation datasets can be found in Appendix E.

| Method | 0-2k | 2-4k | 4-8k | 8k+ | average |
|---|---|---|---|---|---|
| Pythia-SFT | **30.54** | - | - | - | - |
| Pythia-No-SFT | 4.41 | - | - | - | - |
| RCC-Ins-Reconstruction | **28.12** | 23.37 | 21.24 | 17.72 | 22.61 |
| RCC-Ins-Human | 25.36 | **25.15** | **23.63** | **20.48** | **23.15** |
| RCC-Ins-Compress | 18.77 | 21.36 | 20.02 | 18.14 | 19.61 |

Table 2: Scores of different models on the task of Document QA.

As shown in Table 2, the fine-tuned pythia-1.4b significantly improved in following instructions. Notably, Pythia-1.4b supports a maximum sequence length of 2048 tokens, so we only used samples under 2k tokens for its evaluation. Our method supports LongBench's maximum input length of 15k tokens within the effective window length of the decoder. We further evaluated the following types for our method:

**RCC-Ins-Reconstruction**, which reconstructs instructions from compressed vectors, scored 28.12 at a length of 2k. It uses the reconstructed instructions to generate responses. This score is comparable with Pythia-sft, demonstrating that RCC can maintain high-quality inference even with a compression ratio of up to $32\times$. This method's average score surpasses that of RCC-Ins-Compress, which compresses both instructions and context, verifying the effectiveness of instruction reconstruction. Due to the fine-tuning dataset being limited to 2k tokens, RCC-Ins-Reconstruction performs poorly in instruction reconstruction when handling longer samples.

**RCC-Ins-Human** directly inputs real instruction texts into the decoder. Compared to the performance fluctuations of RCC-Ins-Reconstruction with increasing sample length, RCC-Ins-Human ex-

hibits more stable performance, especially maintaining efficient inference at lengths beyond 8k. We attribute this to the decline in instruction reconstruction quality in RCC-Ins-Reconstruction for long texts, whereas RCC-Ins-Human employs fixed instructions, unaffected by length.

**RCC-Ins-Compress** compresses both context and instructions simultaneously. The encoder receives concatenated texts, and the decoder is only prompted with brief information, such as "Answer is:". This strategy's limited ability to understand instructions and context gives it a low average score of 19.61. It particularly underperforms compared to RCC-Ins-Human and RCC-Ins-Reconstruction in samples under 8k. However, with ultra-long samples (8k+), its performance becomes comparable to RCC-Ins-Reconstruction. This is likely due to the latter's deficiencies in instruction reconstruction at extreme lengths. Specific model generation results can be found in Appendix E.

### 4.3.4 More Powerful Base Model

To delve into the potential of RCC, we have employed the LLaMA2-7B (Touvron et al., 2023b) model, which possesses a larger parameter size, as both the encoder and decoder for RCC. In order to accommodate the constraints of training resources, we utilized the LoRA (Hu et al., 2021) approach to efficiently fine-tune the LLaMA2-7B model. Table 3 presents the experimental results, which indicate that the more powerful LLaMA2-7B model outperforms the Pythia-1.4b model. As the text length exceeds 8k, the performance of the Pythia-1.4b model significantly degrades, whereas the LLaMA2-7B model remains less affected. This outcome demonstrates that RCC can be readily scaled to more robust base models, thereby achieving superior performance.

| Length | RCC-Pythia-1.4b | RCC-LLaMA2-7B | $\Delta$ |
|--------|-----------------|---------------|----------|
| 0-2k | 32.15 | 44.68 | +12.53 |
| 2-4k | 33.11 | 35.07 | +1.96 |
| 4-8k | 29.96 | 31.15 | +1.19 |
| 8k+ | 23.38 | 30.78 | +7.4 |

Table 3: The table presents the average scores of the 1.4b and 7b models across various text lengths on the 2wikimqa and HotpotQA datasets. The results indicate a clear performance superiority of the 7b model over the 1.4b model, particularly in the 0-2k and 8k+ text length categories. It is worth noting that both models were fine-tuned using data from the training set of HotpotQA.

| Method | Context Length (MiB) | | | | | | |
|--------|------|-------|-------|-------|-------|--------|--------|
| | 0 | 2k | 16k | 64k | 92k | 1024k | 1350k |
| CEPE | 18000 | 18300 | 20888 | 46968 | 78894 | – | – |
| RCC | 18198 | 18694 | 19302 | 21180 | 22058 | 55858 | 75686 |

Table 4: The table compares the memory consumption of RCC and CEPE when processing long texts. CEPE uses an encoder with 0.4B parameters and cross-attention weights of 1.4B. RCC uses the Llama model with 1.8B parameters as the encoder. Both RCC and CEPE use LLaMA2-7B as the decoder.

### 4.3.5 Memory Usage Analysis

In Figure 1, we observe when RCC processes text up to 2k tokens, the GPU memory usage only increases by approximately 0.1 GB. In contrast, the original Pythia-1.4b experiences a 2 GB increase in memory usage for 2k token. When processing 16k token text, The total memory usage of Pythia-1.4b will be double that of RCC. Since we use a compression rate of 32 $\times$, RCC can save up to nearly 32 $\times$ in storage space as text length increases. We also conducted a comparative analysis of the CEPE model, which is another approach aimed at extending the context window of LLMs. The CEPE model consists of an encoder and a decoder. To facilitate efficient inference of long contexts, the CEPE model has significantly reduced the number of parameters in its encoder, while maintaining a substantial parameter count in its cross-attention and decoder modules. In contrast to the RCC architecture, the CEPE (Yen et al., 2024) encoder does not compress the context length, leading to a significantly shorter context length that can be processed within the same memory

limitations. This is illustrated in Table 4, where CEPE's capacity for handling context length is shown to be inferior to that of the RCC architecture. More information on additional memory consumption can be found in Appendix A.

## 5 CONCLUSION

Utilizing compression techniques to mitigate challenges in long-text training and inference has proven to be a highly promising strategy. Our work quantitatively analyzes the impact of different context lengths on compression performance. It also achieves higher compression rates than previous methods, significantly enhancing the ability of LLM to handle long texts. To address the substantial resource consumption of long-text training, we proposed a two staged training strategy that further improved the efficiency of the model in processing long-text training. RCC model demonstrated outstanding performance across multiple tasks, including context compression reconstruction, long-document question answering, and passkey retrieval tasks with sequences up to 1 million tokens. Additionally, we analyzed the issues arising from simultaneous compression of context and instructions and propose an instruction reconstruction method that effectively alleviated these problems.

The RCC method has significantly advanced text compression efficiency. It has also improved long-document question answering, but it has limitations. Although the parameter count of RCC's encoder and decoder is twice that of Pythia, the impact of the model's parameter count on storage space significantly diminishes with increased text length. Additionally, the lack of long-text instruction fine-tuning data has caused performance bottlenecks for RCC. Our experiments show the critical impact of training data on the model's performance. The effectiveness of language models fine-tuned with instructions depends largely on the quality and coverage of those instructions. These issues provide clear directions for our future research.

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

## A GPU MEMORY CONSUMPTION ANALYSIS

We ran the model on an A800 GPU using HuggingFace's Transformers library (Wolf et al., 2020) and tested the GPU memory consumption of different models. As shown in Figure 5, the GPU memory usage of Pythia-1.4b increases rapidly with the length of the input context. When the context window reaches 64k, the model's GPU memory usage exceeds 60GB. RCC-1.4 x 2b, where both the encoder and decoder are Pythia-1.4b models with a compression rate of 32, shows that when its GPU memory usage exceeds 60GB, it processes a context length close to 2048k tokens. This is 30 times the length Pythia-1.4b can handle, nearly matching the compression rate.

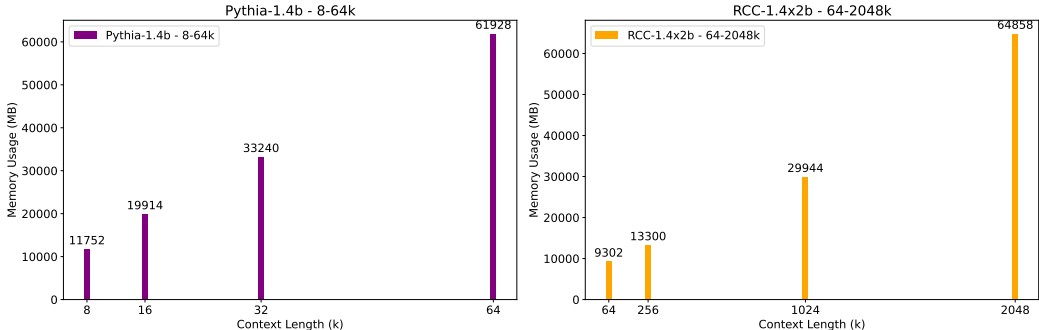

Figure 5: When the GPU memory approaches 60GB, the memory occupation of different models. Left: Pythia-1.4b, Right: RCC model using Pythia-1.4b for both encoder and decoder. Both models utilize FlashAttention-2 (Dao, 2023).

## B RANDOM PROMPT TEXT RECONSTRUCTION TASKS

The random prompt text reconstruction tasks involves an original text sequence $(w_1, \ldots, w_n)$, where the encoder compresses the entire original text and produces a compressed hidden vector $(H)$. The decoder then needs to reconstruct the text that follows the random prompt word in the original text based on the encoder's input vector and the random prompt word from a segment of the original text. The prompt word is a substring of the original text $(w_i, \ldots, w_j)$, denoted as $p$, and the target sentence to be reconstructed, which follows the prompt word, is $(w_j, \ldots, w_x)$, denoted as $c$.

$$\mathcal{L}_{\text{RAE}} = \max_{h,\ldots,p} P\left(\boldsymbol{c} \mid h, \ldots, p; \Theta_{LLM}\right)$$

In the text continuation task, the prompt is no longer a substring of the encoder's input text but is the immediately following segment of text, and the target sentence is still the text that comes right after the prompt. The formula for the text continuation task is the same as that for the random prompt text reconstruction task.

## C  FORMAT OF PASSKEY RETRIEVAL

We follow the text format for passkey retrieval from existing works (Chen et al., 2024; Mohtashami & Jaggi, 2023). The format of the document is as follows:

```
There is an important info hidden inside a lot of irrelevant text.
Find it and memorize them.  I will quiz you about the important
information there.

The grass is green.  The sky is blue.  The sun is yellow.  Here we
go.  There and back again.  (repeat M times)

The pass key is 56994.  Remember it.  56994 is the pass key.   The
grass is green.  The sky is blue.  The sun is yellow.  Here we go.
There and back again. (repeat N times)

What is the pass key?  The pass key is
```

## D  EFFECTS OF TEXT RECONSTRUCTION

The example of our method's reconstruction effect at $32\times$ compression rate is shown below. As the table 5 indicates, our method has almost completely reconstructed the context.

## E  FINE-TUNING DATASETS AND MODEL-GENERATED CASES

Table 6 displays information such as the sources, average lengths, and computational metrics for various tasks. Below is a sample data entry for document question answering, primarily consisting of three parts: '*input*', '*context*', and '*answers*'. The '*input*' represents the prompt or instruction, the '*context*' is the surrounding text the model needs to search through, which is often lengthy, and the '*answers*' represent the possible answers derived from the context. Example:

```
input:  "Which park is further south within Spain, Picos de Europa
National Park or Timanfaya National Park?"

context:  'Passage 1:Lake Ercina Lake Ercina is a small highland
lake ...  The population is 47 (INE 2016).'

answers:  ['Timanfaya National Park']
```

When using RCC-Ins-Reconstruction for instruction reconstruction inference, we concatenate the '*context*' and '*input*' parts of the sample with a newline character and input them into the model's encoder for compression. Simultaneously, the decoder's input is a fixed prompt:

```
prompt:  "system:  You are a helpful assistant.  user:   "
```

The decoder, starting with this prompt, first reconstructs the instruction and then answers the question based on it. The content generated by the model is shown in blue font:

```
"system:  You are a helpful assistant.  user:  Which park is
further south within Spain, Picos de Europa National Park or
Timanfaya National Park?  assistant:  Timanfaya National Park"
```

The model accurately reconstructed the instruction and provided the correct answer, '*Timanfaya National Park*'.

| Our Result on RCC-32-Transformer | Standard Result |
|---|---|
| The Access nodes and storage daemons make up a data plane, while the core provides its control plane. Also: How IBM Watson is revolutionizing 10 industries TechRepublic So, what does all mean for customers? Itś multi-cloud storage management, which enables allows you to manage, deploy, and migrate data storage across private and major public clouds. This includes Alibaba, AWS, Azure, and Google Cloud. Itś easy to see why Red Hat values this. It gives their customers a way to manage storage without sweating the details across multiple platforms. As Ranga Rangachari, Red Hatś vice president of Storage and Hyperconverged Infrastructure, said in a statement: "Data portability is a key imperative for organizations building and deploying cloud-native applications across private and multiple clouds. NooBaaś technologies will augment our portfolio and strengthen our ability to meet the needs of developers in todayś hybrid and multicloud world. We are thrilled to welcome a technical team of nine to the Red Hat family as we work together to further solidify Red Hat as a leading provider of open hybrid-cloud technologies." Related stories: Kidderminster-based Renault UK Clio Cup ace Dan Rowbottom will join Ciceley Motorsport for the 2019 British Touring Car Championship. Backed by Cataclean, the lead valuable additive to clean and fuel engine restore and exhaust systems, Rowbottom will graduate from the Renault UK Clio Cup into one of Ciceley's Mercedes-Benz A-Class cars for the forthcoming campaign. He was a triple race winner last season his way to fourth place | The Access nodes and storage daemons make up a data plane, while the core provides its control plane. Also: How IBM Watson is revolutionizing 10 industries TechRepublic So, what does all mean for customers? Itś multi-cloud storage management, which enables allows you to manage, deploy, and migrate data storage across private and major public clouds. This includes Alibaba, AWS, Azure, and Google Cloud. Itś easy to see why Red Hat values this. It gives their customers a way to manage storage without sweating the details across multiple platforms. As Ranga Rangachari, Red Hatś vice president of Storage and Hyperconverged Infrastructure, said in a statement: "Data portability is a key imperative for organizations building and deploying cloud-native applications across private and multiple clouds. NooBaaś technologies will augment our portfolio and strengthen our ability to meet the needs of developers in todayś hybrid and multicloud world. We are thrilled to welcome a technical team of nine to the Red Hat family as we work together to further solidify Red Hat as a leading provider of open hybrid-cloud technologies." Related stories: Kidderminster-based Renault UK Clio Cup ace Dan Rowbottom will join Ciceley Motorsport for the 2019 British Touring Car Championship. Backed by Cataclean, the leading fuel additive to clean and restore engine fuel and exhaust systems, Rowbottom will graduate from the Renault UK Clio Cup into one of Ciceley's Mercedes-Benz A-Class cars for the forthcoming campaign. He was a triple race winner last season his way to fourth place |

Table 5: Random prompt text reconstruction case.

| Dataset | Task | Source | Avg len | Metric | count |
|---|---|---|---|---|---|
| Qasper | Single-Document QA | Science | 4,620 | F1 | 224 |
| MultiFieldQA | Single-Document QA | Multi-field | 4,558 | F1 | 150 |
| HotpotQA | Multi-Doc QA | Wikipedia | 6,657 | F1 | 300 |
| 2WikiMultihopQA | Multi-Doc QA | Wikipedia | 6,146 | F1 | 300 |

Table 6: LongBench-E Information

Additionally, we tested the RCC-Ins-compress model. The input to the RCC-Ins-compress encoder is identical to that of the RCC-Ins-Reconstruction, but the decoder's prompt is:

```
Prompt:  "Response of system:"
```

Since RCC-Ins-compress has not been trained on instruction reconstruction tasks, it does not reconstruct the instruction in its output. Instead, it directly answers the question based on the mixed compressed context and instruction, which may result in the model failing to follow the instruction.The content generated by the model is shown in blue font:

```
"Response of system:  Panic of 1797"
```

It can be seen that the model made an error in following the instructions.

