# OpenReview forum: "Recurrent Context Compression: Efficiently Expanding the Context Window of LLM"
_ICLR.cc/2025/Conference — Submitted to ICLR 2025_

### Official Review · Reviewer_t2N7 · 2024-10-24

**Soundness:** 2
**Presentation:** 3
**Contribution:** 2
**Rating:** 3
**Confidence:** 3

**Summary:**

This paper proposes Recurrent Context Compression, or RCC for short, that is designed to extend the context length of large language models. RCC can be trained with both text reconstruction and text continuation objectives, and a special instruction reconstruction process is used to mitigate the issue of performance degradation after compression. RCC is tested across multiple tasks and achieves comparable performance to un-compressed methods.

**Strengths:**

S1. RCC achieves a high compression ratio compared to its baseline method ICAE and is proven to preserve most of the context information across experiments of text reconstruction and downstream tasks.

S2. The training paradigm of RCC is carefully designed. For example, it proposes an instruction-aware reconstruction task to mitigate the performance degradation issue. Experiments show that the instruction reconstruction is taking effect in the ability of LLMs to follow the instructions. It also proposes a short-to-long training paradigm to save the computational overhead during training.

**Weaknesses:**

Some ideas proposed in this paper are not new:

W1: The paper improves the final-layer compression solution of ICAE (Ge et al., 2024) and AutoCompressor (Chevalier et al., 2023) by incorporating the hidden representation of each layer. However, similar ideas are used by prior work such as Compressive Transformers (Rae et al., 2020) and Dodo (Qin et al., 2024). The solution adopted by RCC is to split and re-use the last token representation of each segment at each layer, while this idea has already been applied by prior work. Compressive Transformers compress each segment with a pooling operator and Dodo dynamically selects the split points instead of employing a fixed-length segmentation, both working at layer level.

W2: Gradual training on from short to long sequences was deployed in many prior works, though for different purposes. E.g., the training sequences of Longformer are gradually prolonged from the limit of BERT to much longer.

Some experiments in this paper are inconclusive:

W3: The paper claims that RCC outperforms ICAE in the task of text reconstruction. However, text reconstruction is a relatively easy task for LLMs and a BLEU score of 0.95 isn't good enough. E.g., in the Dodo paper, LLaMA can achieve a BLEU of 0.98 with a compression ratio of 20x.

W4: Many experiments only have infini-transformers as the baseline. Comparing to a single baseline is not sufficient to support that RCC is superior to prior methods. Some related methods, such as Recurrent Memory Transformer (RMT), should be tested.

Other comments:

W5: RCC can compress the context after the pre-filling. However, the major computational overhead of the transformer decoding is not the attention but the non-quadratic terms according to the estimation of this blogpost. This effect can be stronger for larger models. It would be great if the authors could show the wallclock time difference between LLM decodings with and without RCC.

References:
- Bulatov, Aydar, Yury Kuratov, and Mikhail Burtsev. "Recurrent memory transformer." Advances in Neural Information Processing Systems 35 (2022): 11079-11091.
- Qin, Guanghui, et al. "Dodo: Dynamic Contextual Compression for Decoder-only LMs." Proceedings of the 62nd Annual Meeting of the Association for Computational Linguistics (Volume 1: Long Papers). 2024.
- Beltagy, Iz, Matthew E. Peters, and Arman Cohan. "Longformer: The long-document transformer." arXiv preprint arXiv:2004.05150 (2020).

**Questions:**

Q1. RCC is trained to re-construct the instruction to prevent the model from forgetting the instruction. However, an alternative approach is to prepend/append the instruction after the compressed representation. As shown in Table 2, RCC-Ins-Human achieves better performance than compressing the instruction. Can you discuss the advantage of compressing the instructions?

---

### Official Review · Reviewer_dDMd · 2024-11-01

**Soundness:** 3
**Presentation:** 2
**Contribution:** 2
**Rating:** 3
**Confidence:** 4

**Summary:**

The study proposes a new KV cache compression method, called Recurrent Context Compression (RCC). The proposed framework consists of a RCC encoder and a RCC decoder. The RCC encoder compresses the input prompt segment-by-segment into a much shorter vector of context. The compressed prompt is then under-gone under a MLP, and fed to the RCC decoder afterward. The framework requires fine-tuning. The proposed method claims significant performance gain on long-context tasks.

**Strengths:**

- The model claims substantial performance improvement on long-context task.
- The idea is straight-forward and easy to understand.

**Weaknesses:**

1. The experiment lacks comparison with other KV cache compression methods & baselines. For example, how is the proposed method compared with other methods on LongBench-E Document tasks or NIAH [1] [2] [3] [4]?
2. What is the memory usage & time efficiency gain in comparison with other methods given the proposed method require fine-tuning to perform prompt compression?
3. The architecture used in the experiment, i.e. pythia-1.4b & mamba-1.4b, is small. I also saw the study did experiment on llama-2-7b in section 4.3. Can the study provide more results on more architectures such as llama3 family or mistral family for the encoder & decoder?
4. If we vary the length of segment to compress in the encoder, I'm wondering how influence can this factor be? Can we do an ablation study to test this?
5. Even though the observation on context-instruction confusion due to prompt compression seems interesting, is there any experimental result to support this claim?

6. Nitpicking: the result presentation part is somehow unclear:
- In Table 1: what  does three numbers in each cell present? For example, "96/98/94"
- In Table 2: is each cell the average result among 4 subtasks from LongBench-E with the specific context length from the column's header?

While the proposed method is interesting, I feel like the experiment is incomplete to show the effectiveness of the method, so I recommend rejection for further improvement for now, but I am willing to reconsider and increase the score a bit if the authors can address my upper concerns.

[1] MInference 1.0: Accelerating Pre-filling for Long-Context LLMs via Dynamic Sparse Attention. Jiang et al., https://arxiv.org/pdf/2407.02490

[2] Razorattention: Efficient kv cache compression through retrieval heads, 2024. Tang et al., https: //arxiv.org/abs/2407.15891

[3] InfLLM: Training-Free Long-Context Extrapolation for LLMs with an Efficient Context Memory, Xiao et al., https://arxiv.org/pdf/2402.04617

[4] KIVI: A Tuning-Free Asymmetric 2bit Quantization for KV Cache, Liu et al., https://arxiv.org/pdf/2402.02750

**Questions:**

Please see weaknesses.

---

### Official Review · Reviewer_bbjH · 2024-11-03

**Soundness:** 2
**Presentation:** 2
**Contribution:** 3
**Rating:** 5
**Confidence:** 4

**Summary:**

The paper studies expanding the context window of the language model by introducing a decoder that chunks the input context into fix size pieces and compressing each. They also consider the effect of prompt and how to compress prompts such that degredation in different forms of prompting is minimized. They evaluate their method on a set of benchmarks and show effectiveness up to 1M length context.

**Strengths:**

- The paper targets a very important problem. Effectively increasing the context of language models is an active area of research and has important practical implications
 - The work is original. Although I have some hesitations about the appraoch, introducing a decoder the way has been presented is novel to the best of my knowledge.

**Weaknesses:**

- The approach proposed basically doubles the number of parameters required for the model. This has both training and inference implications.
- As shown in Table 2, the method results in 2 point degradation for the context length that is within the pretraining setting. This is significant and further evaluations and studies is needed. One of the main challenges of long context extension is preserving short context benchmarks.

**Questions:**

- For the decoder part, have the authors considered a model that uses a bi-directional attention mask?
- Have the authors compared the method to models such as T5 with iso parameters (decoder + encoder)

---

### Official Review · Reviewer_Jqs9 · 2024-11-04

**Soundness:** 2
**Presentation:** 3
**Contribution:** 2
**Rating:** 5
**Confidence:** 4

**Summary:**

The paper introduces Recurrent Context Compression (RCC), a technique for long-context language modeling. The input is segmented into smaller chunks, the encoder encodes multiple chunks at a time, and the hidden state of the last token in each segment at a layer is fed into the decoder. On the decoder side, the hidden states from the first layer are prepended to the decoder input tokens and the hidden states from the encoder are connected to the hidden states in the decoder at each layer.
The paper also introduces reconstruction tasks to help the decoder model to enhance memory and retrieval ability.
The method is evaluated on the task reconstruction task, passkey retrieval, and LongBench.

**Strengths:**

- The method is well-motivated—building long-context language models is a challenge and useful task. The method is designed to tackle the training and inference challenges and achieves better memory usage compared to some previous work.
- The paper includes experiments using different settings of RCC, such as different encoder and decoder models, and different ways of using the instructions during training.
- RCC achieves strong PassKey results at 1M input tokens in supervised settings, which suggests the architecture may be capable of passing such synthetic stress tests.

**Weaknesses:**

- In RCC, the question and the instruction are first reconstructed from the encoder hidden states. However, this means that these tokens are still inputted to the decoder, and the time and space complexity is worse than simply putting these tokens in the decoder inputs in the first place.
- The evaluation of downstream applications is limited to only the QA subsets from LongBench, and the comparison is limited to only Pythia with and without fine-tuning (Table 2). Similarly, the passkey experiments (Table 1) are limited in comparisons with previous related long-context language modeling works, such as AutoCompressor (Chevalier et al., 2023), LLoCO (Tan et al., 2024), Unlimiformer (Bertsch et al., 2023), and StreamingLLM (Xiao et al., 2024). Due to the lack of comparisons, it is unclear how RCC compares to or improves upon previous methods.
- Overall, there are still some issues with the evaluation settings (detailed below in the Questions section), but the work could really benefit from testing RCC on more downstream applications as well as on out-of-domain tasks.

**Questions:**

- What are the results of RCC on more out-of-domain tasks other than QA tasks? I would be interested in seeing the results of LongBench in other categories like summarization and few-shot learning.
- How do you evaluate LongBench at 0-2k? Do you only use the test samples where the input length is in that range or do you do truncation?
- Similarly, what is the performance on the passkey retrieval tasks without specifically fine-tuning on the task? In previous works (Fu et al., 2024, Dubey et al., 2024), the synthetic tasks are often evaluated in a zero-shot setting without doing any fine-tuning as a stress test, it would reveal more about the method if it were only trained on the Pile and test on the synthetic tasks.

---

### Meta-Review · Area_Chair_JByM · 2024-12-20

**Metareview:**

Based on the reviewers' feedback, I recommend not to accept this paper at this time. While the Recurrent Context Compression (RCC) approach presents an interesting direction for extending LLM context length, several critical issues remain: limited novelty compared to prior work like Compressive Transformers and Dodo, insufficient experimental comparisons beyond Infini-transformers, and concerns about computational overhead given RCC doubles the parameter count. The experimental results are also inconclusive, with text reconstruction performance below existing methods. Without author response to these concerns, the paper's contributions remain inadequately demonstrated.

**Additional Comments On Reviewer Discussion:**

I have read the messages in the discussion period and my opinion has been summarized as in the metareview above. I considered these points in my recommendation.

---

### Decision · Program_Chairs · 2025-01-22

Reject